# Psychometric Properties (Measurement Invariance and Latent Mean Difference across Gender) of the Learning Transfer Inventory System (LTIS) to Assess Thai Teachers’ Learning Transfer of COVID-19 Prevention Guidance in the Classroom

**DOI:** 10.3390/ijerph19159439

**Published:** 2022-08-01

**Authors:** Bekolo Ngoa Celestin, Tchouchu Emmanuel, Ekoto Eugene Christian, Surapong Ngamsom, Kouame Dangui Dorcas, Agyemang Rama

**Affiliations:** 1Department of Economics and Business Administration, Yibin University, No. 8, St. Luke, Wuliangye, Yibin 644000, China; 2Department of Management Studies, School of Management, University of Cape Coast, Cape Coast 00233, Ghana; temmanuel@ucc.edu.gh; 3Department of Management Studies, Adventist University of Haiti, Carrefour HT6131, Haiti; dr.ekoto1976@gmail.com; 4English for Integrated Study (EIS) Association, Rayong 21000, Thailand; mreis@eisthe.org; 5School of Management and Economics, University of Electronic Science and Technology of China (UESTC), No. 2006, Xiyuan Ave, West Hi-Tech Zone, Chengdu 611731, China; koudordan@yahoo.com (K.D.D.); rmohammedan@gmail.com (A.R.)

**Keywords:** learning transfer systems inventory (LTSI), construct validation, confirmatory factor analysis (CFA), measurement invariance, latent mean difference

## Abstract

One of the most important ways to improve, update, and sustain teachers’ skills in an institution is via training. Nonetheless, despite the resources invested in training, learners’ mobilization of new learning after they return to work does not always reach expectations, in part because of a lack of learning transfer assessment tools. This study investigated the psychometric properties of the learning transfer inventory system (LTSI) in assessing the teachers’ transfer of COVID-19 prevention measures in Thai public school institutions. Participants were a sample of 700 in-service teachers (females = 54.8%; mean age = 36 years, SD = 15.41) who completed training on health code guidance for COVID-19 prevention in school. Results following confirmatory factor analysis, a test of the measurement invariance and measurement of the latent mean difference across gender, of the instrument yielded support for the hypothesized 16-factor structure. Empirical support for discriminant and convergent validity was strong. Additionally, we found a significant latent mean difference between male and female teachers related to the constructs peer support, supervisor sanction, and training design. The LTSI appears to yield valid and reliable scores for measuring the learning transfer of Thai teachers following in-service training.

## 1. Introduction

The importance of teachers’ in-service training for COVID-19 prevention guidance at the school setting is well established for students and community safety [1,2]. Teacher in-service training for epidemic control equips teachers with the resources, skills, and assistance they need to properly carry out their profession’s tasks, inclusive of many health procedures, policies, and regulations recently integrated in school administration [3,4]. The merits of in-service training depend on learning transfer to the work setting (classroom) [5]. Learning transfer was first defined as “the degree to which participants apply the knowledge, skills and attitudes acquired in training in their professional activity” [6].

Although health guidance trainings have the potential to scale up teachers’ ability to prevent virus contamination in school [7,8], much less research has investigated factors that exhibit or inhibited such abilities to transfer in the classroom setting because of a lack of a better learning transfer assessment tool [9,10]. Past measurements of the scale of learning transfer were found to contain some flaws due to their inability to identify and evaluate all possible factors that influence learning and learning transfer both during training program itself and at the workplace [11] Further, in an epidemic situation, the lack of a cross-culturally valid tool to quantify learning transfer predictors has been the biggest impediment to health educators, school administrators, and government official to harmonizing strategies and replicating good practices [12]. Such tools are also needed to assess the return on investment from teachers’ training based on learning transfer at the workplace.

In the literature, many studies have highlighted the influence of contextual parameters on the measurement of learning transfer [13,14,15] more specifically with regard to the administration context of the measure (e.g., source of transfer evaluation, time of transfer measurement, etc.) and parameters of the training itself (e.g., duration of training, type of skills developed, pedagogical strategies employed). According to Taylor et al. [16], the source of transfer evaluation would influence the evaluation in such a way that it would be higher when coming from the learner (self-reported measure) as compared to an evaluation coming from another source (p e.g., immediate superior or colleagues). In the same vein, Baldwin et al. [17] argued that there would be a significant gap between the level of transfer anticipated by learners and that achieved. In general, findings suggest that the benefit of workplace learning transfer varies from the modest 40% of training knowledge transfer six months after training [16] to a low 10% rate of training transfer after a year [18]. Other studies have reported that certain behavioral changes can sometimes take up to a year to appear, in particular due to constraints present in the workplace or specific characteristics of the behaviors being transferred [19]. Thus, in addition to the characteristics of the training already recognized to promote the transfer of learning (e.g., relevance of learning objectives, relevance of training content, activities for putting learning into practice during training) [13,14,15] added new characteristics relating to the current context of organizations, such as the mode of dissemination (formal vs. informal; self-learning, peer learning, or classroom learning, etc.) or even the trainer himself [17]. Comprehensive measurement tools would be helpful; first, for the cost–benefit analysis of in-service training such as epidemic prevention training, and, second, for understanding the key factors that enhance learning transfer to actively promote those factors and, at the same time, work toward eliminating inhibitive barriers against transfer. Our study aimed to validate a workplace learning-transfer measure among Thai school teachers.

In developing countries such as Thailand, there is a keen interest in upskilling teachers using in-service training [9] and in the evidence base for the benefit to work performance from the training [12,20]. In the past, the aim of in-service training was to address issues related to group heterogeneity among teachers (e.g., various development stages; unskilled, mechanical, routine, and professional stages) and to give regular updates to changes in curriculum and pedagogy [21]. Recently, the COVID-19 pandemic has generated a significant scale of transformation in the educational system, which requires updated knowledge to be transferred in the classroom [22]. In Thailand, however, evidence for such transfer from the workshop to the workplace has yet to be evaluated [23,24]. Little is known about the factors associated with the transfer of learning, both in the training setting and in the physical environment of the classroom after training.

Further, assessing the findings and analyzing the variables that have an impact on employees’ learning transfer are often complicated when the biographical characteristics of the employee are taken into consideration [25]. For instance, even though there are no significant variances among men and women in their problem-solving capabilities, critical skills, competitive drive, motivation, learning ability, or friendliness, women are said to be more conforming and have lower achievement expectations than males [25]. In general, when it comes to training participation, several researchers have found that females have a more positive attitude than males [16,26]. Thus, their perception of learning transfer may differ. In the context of a female-dominated sector like education, this study sought to validate a widely used measure of a learning transfer inventory system (LTIS) in a sample of Thai teachers and to test for measurement invariance and the latent mean difference of the instrument across gender.

While Thailand is grappling with the third wave of the COVID-19 epidemic, teachers’ lack of evidence for the workplace transfer of learning after in-service training on an epidemic control program is concerning for the Ministry of Education (MOE) in Thailand [27]. The goal of LTIS instruments in a local school setting is to offer evidence for school principals to proactively address problems that prevent worker to transfer the trained skills, such as health code guidance for epidemic prevention. In such a context, indicators of the transfer of learning include: (1) the ability of learners to apply their learning in work settings that are more or less comparable to those provided in the context of the training (“generalization of knowledge”) [28] and (2) the ability to apply their new learning outcomes over time, not just in the immediate aftermath of the training (maintenance over time) [5]. To assess such indicators, according to the meta-analysis of Blume et al. [14], factors that influence the transfer of learning can span from (1) individual variables (e.g., self-efficacy, motivation, attitudes), (2) variables relating to training (e.g., training design), and (3) the work environment (e.g., resistance to change). In addition, the perceived usefulness of the training or the value that the learner attributes to his/her participation in the training would be relevant to the transfer of learning to the actual workplace [16]. Similarly, environmental elements, such as organizational culture [16] and the support of peers [29] or that provided by the immediate superior [30], are extremely important to impacting the transfer process [20]. 

Among the measures of learning transfer, the learning transfer inventory system (LTIS, Holton, et al. [31]) is among the most widely used methods for evaluating training transfer across organizations, training types, and multiple cultural contexts. The LTIS measures training transfer in two dimensions including (1) 11 factors for training specific domain (e.g., learning readiness, motivation to transfer) and (2) 5 factors for training general domain (e.g., resistance/openness to change, performance, self-efficacy) [32,33]. In the current COVID-19 pandemic, the LTSI can assess the effectiveness of the measures for epidemic control implemented in the school setting after training to safely keep schools open. It comprehensively identifies and maps the evidence of training transfer both in the training session and the classroom setting and therefore establishes room for improvement. With previous cross cultural validation in the United States [32], France [34], Germany [35], New Zealand [36], Saudi Arabia [37], and South Korea [33] but less so in developing countries, the LTSI is a reliable diagnostic tool to conduct comparative learning transfer research across cultural boundaries. Empirically, convergent, divergent, predictive, and criterion-related validity findings have been discovered using both the original English and translated versions of the instrument in some studies, supporting the instrument’s psychometric soundness [32,33]. Thus, from a theoretical perspective, the LTSI validation in Thailand represents the possibility of identifying a nomological network of the learning transfer system. Some findings on one hand supported the instrument’s 16-factor structure [29,32,33], and other studies have reported a 12-factor model of the LTSI [38], calling for further studies to clarify the structure of the LTSI. There may be gendered differences in learning transfer yet to be assessed in the use of the LTSI, given the fact that in female-dominant fields such as education, gender difference has been shown to lead to difference perception in training transfer [39]. Validation of the utilized latent trait scores may help resolve the inconsistencies in the LTSI factorial structure, and no studies examining the LTSI’s measurement invariance across gender have been conducted (Bates et al., 2012). 

### Goal of the Study

We validated the LTSI in the sample of Thai high school teachers. In doing so, we hypothesized that, with regard to the scores from the LTSI by teachers: 

**Hypothesis** **1** **(H1).**
*The LTIS shows acceptable construct validity.*


**Hypothesis** **2** **(H2).**
*The LTIS shows acceptable discriminant validity.*


**Hypothesis** **3** **(H3).**
*The LTIS shows acceptable convergent validity.*


**Hypothesis** **4** **(H4).**
*The LTIS yields reliable scores.*


## 2. Materials and Methods

### 2.1. Participants and Sample

We recruited a non-probabability convenience sample of 315 men (45.2%) and 384 women (54.8%), a majority (74.6%) in their twenties, thirties, or forties, with service experience of 5 to 25 years. About 43.8 percent of respondents (*n* = 306) had a four-year university degree. In terms of training categories, all teachers participated in the 5-day in-service training for COVID-19 prevention and control guidance.

### 2.2. Measures

LTSI: The LTSI consists of 48 questions to measure 16 training transfer variables in two dimensions: training on specific and general domains (see Table 1). Examples of items for the training on specific domains include: “Do learners feel better able to perform?” and “Do they plan to use their knowledge and expertise?” to access the motivation to transfer learning. Examples of items for the training on general domains include: “Do work group actively resist change?” and “Are they willing to invest energy to change?” to measure openness to change. Items are scored on a 5-point Likert-type scale: 1 (strongly disagree) to 5 (strongly agree). Higher scores mean it is a very important learning transfer factor. 

The control variables were collected as participants reported their (1) gender, (2) age, (3) education level, (4) work experience, (5) job position, (6) school type, and (7) type of training.

### 2.3. Procedure

The Yibin University Economics and Business Administration Ethics Committee approved the study. First, the research project obtained approval from every high school principal before the training program. Second, after the training session, participants were provided information (1) on the research’s objective and nature, (2) that participation was voluntary, and (3) that they had the right to abstain from participating in the study at any moment during the study without penalty. No personal information was requested. The survey was taken online a month after the training or any time soon after.

### 2.4. Data Analysis

Given that the data for this study was collected through an online survey that collected data for both predictors and outcome variables, systematic response bias might result, inflating or deflating the replies. We ran a follow-up analysis to address the issue of common technique bias. First, we looked at Harman’s one-factor approach, which put all 33 specific training elements into a single CMV factor. The components that emerged explained 25.7% of the variation, which is less than 50%, demonstrating that common technique bias was not an issue in this study. Harman’s single-factor test, on the other hand, is known to be extremely cautious in identifying CMB [40]. We used a common method bias test to further investigate the topic of common method bias. The zero constrained model was validated with the chi-square test. As a result, a bias distribution test was performed (of equal constraints). In this test, the chi-square test is significant, suggesting that a test of equal specific bias revealed unevenly distributed bias.

We then conducted exploratory factor analysis (EFA) employing common factors analysis (principal factor axis) with oblique rotation and reliability analysis were performed, and confirmatory factor analysis (CFA) was used to corroborate the factor structures that emerged from the exploratory methods. For EFA, we used Statistical Package for the Social Sciences Version 24 (SPSS24) (IBM, Armonk, NY, USA) and analysis of moments structures (AMOS) to find the latent variables of LTSI comprising training in the specific and general domains in order to achieve the study goals. The LTSI scale’s factor structure was verified using five criteria: (1) a priori the goal was to extract the same number of components as the original authors Bates et al. [32], (2) the proportion of variance explained was set to 60% or higher, (3) the magnitude of factor loadings may be equal to or more than 0.40, (4) a basic structure (no cross loading) was necessary, as well as (5) at least three variables per component to discover stable common factors. We also looked at the communality of each variable to see how much valid variance was explained by each variable’s factor solution as suggested by Hair et al. [41]. Model fit was assessed using (1) chi-square (to assess the difference between the predicted and observed model), (2) the Tucker–Lewis index (TLI) and comparative fit index (CFI) (to measure the amount of shared variance and covariance between the predicted and observed models), and (3) standardized root mean residual (SRMR) and root mean square error of approximation (RMSEA) (to measure the amount of the shared variance and covariance between the predicted and observed models) (to estimate the approximate amount of error variance between the two models). TLI and CFI cut-offs should be larger than 0.90, whereas the RMSEA and SRMR thresholds should be less than 0.06 and 0.08, respectively. In addition, we tested for common method bias, reliability, and validity across the full sample (*n =* 700). To generate a more cautious estimate of convergent validity, the average variance extracted (AVE) was used. It was deemed acceptable if the AVE was equal to or more than 0.50.

Finally, we used a configural invariant test to build a baseline model across groups, following Ployhart and Vandenberg’s [42] technique of testing measurement invariance using tighter steps. Factor loading, intercepts, and residuals are all estimated freely by the configural invariant. This suggests that all groups have the same mental framework [41]. The measurement invariance test is discontinued if the data does not support the configural test, which is a fundamental part of this test. 

In order to measure the metric invariance model, all factors’ loading were required to be the same. This is a test with a low level of volatility. It reveals that the different groups had the same reaction to the indicators. After that, the scalar invariance model was examined, because it is required before latent group comparisons can be made across groups. This test’s process comprises restricting factor loading and indicator intercepts to be the same across groups. The changing value of the CFI was observed in order to determine an appropriate metric and scalar invariance. If the CFI change is more than 0.001, the absence of evidence for metric and scalar invariance is determined. At the end, we initially limited the latent mean of the male group with regard to gender to examine the latent mean differences between genders. The latent mean of female (gender) was then estimated freely. The value of critical ratio (CR) was utilized to evaluate the latent mean difference in all of the above situations, with CR > 1.96 as a threshold for a substantial difference in latent means. For the reliability (Cronbach’s alphas) interpretation, we followed Cohen et al.’s [43] guidelines: (1) excellent (α ≥ 0.90), (2) very good (0.85 ≤ α ≥ 0.90), (3) good (0.80 ≤α ≥ 0.85), and (4) acceptable (0.75 ≤ α ≥ 0.80).

## 3. Results

### 3.1. Exploratory Factors Analysis of LTSI

The latent variables of LTSI comprising training specifics and the general domain were identified using exploratory factor analysis (EFA), employing the principal factor axis with oblique rotation.


Training in the Specific Domain


The Bartlett’s test of sphericity was significant, with $x^{2}$(528) = 9874.712, p<0.00, suggesting that the variables were sufficiently correlated to proceed with the research (see Table 2). The MSA index was 0.732, suggesting that the data was appropriate for EFA. The approach yielded an EFA with no cross loading that was identical to Bates, Holton, and c and Kim et al. [33] (see Table 2). The total factor explaination was 77.9%, which met the threshold for the factor extraction of 60% of the variance. The 33 items’ factor loadings varied from 4.42 to 9.91, exceeding the minimum of 4. There was no evidence of cross.


Training in the General Domain


We analyzed 15 items for the general domain using the same extraction process as for the training in the specific domain, with the number of factors to be extracted set at 5 (see Table 3). The MSA index was 0.875, indicating that the provided data were acceptable for EFA. Bartlett’s sphericity test was significant (χ^2^ = 3500.431, df = 105, *p* = 0.000). The 5 extracted components accounted for 70.27 percent of the total variance, fulfilling the extraction requirement of 60% variance (See Table 3). The communalities varied from 0.44 transfer effort performance expectation (TEPE) to 0.99 resistance and openness to change (ROC1).

### 3.2. Confirmatory Factors Analysis of LTSI


Training in the specific domain


The 33 items resulting from the EFA were classified into 11 constructs in the hypothesized 11-factor model of the training in the specific domain of LTSI. The 11-factor model was an excellent match for the data, according to the CFA(a) findings. The TLI (0.935) and CFI (0.955) values were higher than Hair et al.’s [41] proposed threshold of 0.90. These index values imply that the obtained data set might explain more than 90% of the model’s variance and covariance. The SRMR (0.05) and RMSEA (0.040) values with a 90% confidence interval (CI) of 0.05 to 0.06 demonstrated the reduced amount of error variance across two models. The standardized factor loadings (i.e., regression weights < 0.001) ranging from 0.57 (POP2) to 1 (OUL1) exceeded the minimum standard of 0.5.


Training in the general domain


TLI = 0.950, CFI = 0.962, SRMR = 0.048, and RMSEA = 0.069 (90 percent CI:0.058–0.081). The CFA (b) findings for the training in the general domain of LTSI Model 1 suggested that the 5-factor model with 15 items matched the data well: TLI = 0.950, CFI = 0.962, SRMR = 0.048, and RMSEA = 0.069 (90 percent CI). Although the two tests were statistically significant (χ^2^ = 212.538, df = 80, *p* < 0.001), these results suggested a well-developed measurement model in terms of the amount of shared variance and the low levels of error variance between the two models. The standardized factor loadings (*p<* 0.001) varied from 0.65 for transfer effort performance expectation (TEPE) to 0.912 for feedback and performance coaching (FPC), indicating that convergent validity was present.

### 3.3. Cross Validation of the Optimal CFA Model in a Different Subsample

After discovering that the optimal structure emerging from the CFA(a) and CFA(b) subsamples collected in the academic year 2020/2021 (*n* = 345) was 11-factor extraction for training in the specific domain and 5-factor extraction for the general domain, a crosscheck of these models was performed to verify model fit in the second subsample collected in the academic year 2021/2022 (*n =* 354). The results of this crosscheck are shown in Table 4. The CFA for 11 factors on training in the specific domain extraction and 5 factors in the general domain extraction demonstrated satisfactory fit, with all indices falling within the acceptable range. Furthermore, the fit measure for both domains remained remarkably consistent across the two subsamples. Factor loading for the training in the specific and general domains, respectively, ranged from 0.525 to 0.952 and 0.867 to 0.993, which was satisfactory. All of the elements were loaded on the latent factors that were planned. We support the LTIS 16-factor-extraction structure after considering the abovementioned findings (fit statistics and factor loadings) for both dimensions of training transfers (training in the specific and general domains).

### 3.4. Cross Validation of Reliability and AVE Validity Analysis of the LTSI 

Over the full sample (*n* = 700), we assessed the reliability and validity of the LTSI 11 training in the specific and 5 in the general domain components (see Table 5 and Table 6). The reliability (Cronbach’s) was assessed using IBM-SPSS 23 using the following general criteria proposed by Cohen et al. [43] to interpret the Cronbach’s: (1) excellent (>0.90), (2) very good (0.85–0.90), (3) good (0.80–0.85), and (4) acceptable (0.75–0.80). According to Hair et al.’s 2010 proposal, the threshold value for construct dependability should be 0.70 or above. To generate a more cautious estimate of convergent validity, the average variance extracted (AVE) was used. An AVE equal or greater than 0.50 was considered acceptable [41]. Table 5 and Table 6 give details of the reliability and AVE validation of the LTSI training in the specific and general domains.

### 3.5. Measurement Invariance

Training in the specific domain 

A multi-group SEM was evaluated using the LTSI scale factors of training in particular domains to establish configural invariance across males and females without restricting equality between the groups. The results of the configural invariance test, as given in Table 7, reveal that the structural patterns are consistent across groups ($\chi^{2}=1$819.833, df = 880, $\chi^{2}\;$= 2.068, CFI = 0.950, RMSEA = 0.056). As a result, the configural model may be used as a baseline against which other constrained models in the invariance hierarchy can be compared. The metric invariance was then tested by restricting the factor loadings to be equal for both male and female respondents. Table 7 also shows the findings of the metric invariance model ($\chi^{2}\;\;$= 1856.101, $\chi^{2}/$df = 913, ($\chi^{2}$ = 2.03, CFI = 0.955, RMSEA = 0.055), which shows a satisfactory model fit. Furthermore, the chi-square difference test yields no significant results, implying that metric invariance may be asserted [44]. Furthermore, we may infer that the results support the metric invariance hypothesis when the value of the change CFI (ΔCFI = 0) is lower than 0.01, as suggested by Eisenhardt et al. [45]. Finally, the intercepts across the two groups were restricted to be invariant, resulting in a scalar invariance test. The scalar invariance model fit indices (β = 1819.833, β = df = 880 =/df = 2.068, CFI = 0.955, RMSEA = 0.56) are shown in Table 7. Furthermore, the chi-square difference between the metric and scalar models is non-significant (*p* > 0.05), and the change value of CFI (ΔCFI = 0) is less than 0.01, implying that scalar invariance is upheld.


Training in the general domain.


A multi-group SEM was evaluated using the LTSI scale factors of training in a particular domain to establish configural invariance across males and females without restricting equality between the groups. The results of the configural invariance test, as given in Table 8, reveal that the structural patterns are consistent across groups ($\chi^{2}$ = 320.336, df = 160, ($\chi^{2}$/df = 2.002, CFI = 0.954, RMSEA = 0.054). As a result, the configural model may be used as a baseline against which other constrained models in the invariance hierarchy can be compared. The metric invariance was then tested by restricting the factor loadings to be equal for both male and female respondents. Table 8 also shows the findings of the metric invariance model ($\chi^{2}$ = 332.557, df = 175, ($\chi^{2}$/df = 1.9, CFI = 0.955, RMSEA = 0.051), which shows a satisfactory model fit. Furthermore, the chi-square difference test is not significant, implying that metric invariance is possible. We may infer that the outcome supports the metric invariance hypothesis because the change in CFI is (ΔCFI = 0) less than 0.01. Finally, the intercepts across the two groups were restricted to be invariant, resulting in a scalar invariance test. The model fit indices of the scalar invariance model ($\chi^{2}$= 332.557, df = 175 ($\chi^{2}$/df = 1.9, CFI = 0.955, RMSEA = 0.955) are shown in Table 8. Furthermore, the chi-square difference between the metric and scalar models is non-significant (*p* > 0.05), and the change value of CFI (ΔCFI = 0) is less than 0.01, implying that scalar invariance is upheld.

### 3.6. Latent Mean Comparison

Table 9 shows the latent mean assessments, which reveals that there are substantial variations in supervisor sanction, peer support, and training design. For supervisor sanction, peer support, and transfer design, the latent means of female-perceived training transfer factors were 0.165, 0.15, and 0.10, respectively, lower than those of men. First, the invariant variance assumption must be met for both male and female groups. The chi-square difference test and change in CFI between the scalar invariance and variance invariance models, as shown in Table 8, indicate that the variance invariance model is supported. The Cohen’s d indices were then calculated using the male and female groups’ common standard deviations. The effect sizes for supervisor sanction (d = 0.13), peer support (d = 0.14), and training design (d = 0.13) are all minor, according to the Cohen’s d indices shown in Table 9.

## 4. Discussion

With respect to the training in the specific domain, 33 items were retained for 11 factors identical to the original version. This includes (1) learner readiness, (2) personal outcomes negative, (3) personal capacity for transfer, (4) peer support, (5) supervisory/manager support, (6) supervisory/manager sanction, (7) motivation to transfer training, (8) transfer design, (9) opportunity to use learning, (10) personal outcomes: positive, and (11) personal outcomes: negative. Previous studies report similar findings [31,32,33,34,35]. In a study done in South Korea, the authors tested the validity of the Korean-translated LTSI across 16 organizations, including education institutions. The sample data collected (753) was split for exploratory and confirmatory factors analysis. The result showed that the LTSI scale is a valid instrument measuring training in 11 specific constructs [33]. Likewise, a study done in Jordan and Germany also resulted in retaining 33 items for 11 constructs for training in the specific domain [35,46].

Regarding training in the general domain, 15 items were retained and grouped into 5 factors also identical with the original version of the LTSI factors. These factors include (1) resistance/openness to change, (2) performance self-efficacy, (3) feedback/performance coaching; (4) transfer effort performance expectancy, and (5) transfer effort performance expectations. These results are in line with previous studies [16,47,48,49]. The application of the LTSI among New Zealand physical education teachers revealed results consistent with the 5 factors of training in the general domain [36]. In a similar study done in South Africa using only an exploratory factor analysis, the five general domain factors were retained as the most-significant work environment indicators of learning transfer [50]. 

Scores from the LTSI had discriminant validity in Thailand. A recent study done by Kim et al. [33] using a procedure similar to the one adopted in this study showed a clear replication of the results obtained in the original study done by Bates, Holton, and Paul [32] and by this study. However, due to overlap in constructs, discriminant validity is not as strong as convergent validity in self-report surveys [51].

The comparisons among demographic participants showed that the Thais’ transfer system was different across gender. In this study, the latent mean difference analyses showed only three of sixteen scales were significantly different between male and female participants. Male respondents rated peer support, supervisor sanction, and training design higher than female respondents, but the effect was small. This gender difference could be explained from the contextual and cultural perspective [21]. In Thailand, the formal training of teachers is mainly handled by the government. As teachers receive insufficient attention in the post-training stage, they bear responsibilities to learn from experienced peers especially in male-dominated teaching subjects like science and mathematics [52]. In the same vein, Thai society propagates the ideology of motherhood, which restricts women’s mobility [53]. Such restriction prevents female teachers from participating in training activities outside school premises and after teaching hours. In addition, in this gendered culture, the ideology shapes different lives for men and women by placing them in different social positions and patterns of expectation [54]. This explains why male teachers are more concerned about supervisor sanction.

## 5. Conclusions

### 5.1. Implication for Practice

Theoretically, this study contributes to the literature on workplace learning and human resource development experts in Thailand by providing a validated questionnaire that can be used in future research. Educational institutions should focus on understanding the key factors that enhance training transfer and actively promote those factors. And, at the same time, they should work toward eliminating inhibitive barriers to transfer. The LTSI was found to be a valid and reliable instrument with which to measure learning transfer from the training setting to the working environment. This result was in line with previous findings [29,37,55,56]. Accurate findings are enabled by proper measurements, which improve intervention efficacy. The LTSI can be used by practitioners and academics researchers alike [33].

### 5.2. Limitations of the Study and Suggestions for Future Research

In terms of sampling strategy, this study has limitations on the generalization of the findings because this study used purposive sampling. Future research should adopt the random sampling process to increase the representativeness of the sample. Moreover, a more heterogeneous sample is recommended in examining the construct validity of the LTSI as well as the differences of the Thai transfer system characteristics. For example, this study gathered the sample from one school type of organizations (public organizations); therefore, future studies should collect data from other school types as well.

## Figures and Tables

**Table 1 ijerph-19-09439-t001:** Learning Transfer System Inventory (LTSI) Scale Descriptions.

Scale Name	Scale Description	Domain
Learner Characteristics Scales		
Learner Readiness	The extent to which individuals are prepared to enter and participate in a training program	Specific
Performance Self-Efficacy	An individual’s general belief that he or she is able to change his or her performance when he or she wants to	General
Motivation Scales		
Motivation to Transfer Learning	The direction, intensity, and persistence of effort toward using in a work setting the skills and knowledge learned in training	Specific
Transfer Effort Performance Expectations	The expectation that effort devoted to transferring learning will lead to changes in job performance	General
Performance-Outcomes Expectations	The expectation that changes in job performance will lead to outcomes valued by the individual	General
Work Environment Scales		
Feedback/Performance Coaching	Formal and informal indicators from an organization about an individual’s job performance	General
Supervisor/Manager Support	The extent to which supervisors/managers support and reinforce the use of training on the job	Specific
Supervisor/Manager Sanction	The extent to which individuals perceive negative responses from supervisors/managers when applying skills learned in training	Specific
Peer Support	The extent to which peers reinforce and support use of learning on the job	Specific
Resistance/Openness to Change	The extent to which prevailing group norms are perceived by individuals to resist or discourage the use of skills and knowledge acquired in training	General
Personal Outcomes: Positive	The degree to which applying training on the job leads to outcomes that are positive for the individual	Specific
Personal Outcomes: Negative	The extent to which individuals believe that not applying skills and knowledge learned in training will lead to negative personal outcomes	specific
Ability Scales		
Opportunity to Use Learning	The extent to which trainees are provided with or obtain resources and tasks on the job enabling them to use training on the job	specific
Personal Capacity for Transfer	The extent to which individuals have the time, energy, and mental space in their work lives to make changes required to transfer learning to the job	specific
Perceived Content Validity	The extent to which the trainees judge training content to accurately reflect job requirements	specific
Transfer Design	The extent to which (a) training has been designed and delivered to give trainees the ability to transfer learning to the job and (b) training instructions match job requirements	specific

**Table 2 ijerph-19-09439-t002:** Rotated Factor Loading Table for the LTSI Training Specific Domain (11 factors extraction *n* = 345).

	Factor										
	1	2	3	4	5	6	7	8	9	10	11
MTL1	0.771										
MTL2	0.915										
MTL3	0.934										
SSA1		0.851									
SSA2		0.930									
SSA3		0.735									
OUL1			0.993								
OUL2			0.881								
OUL3			0.951								
LR1				0.871							
LR2				0.921							
LR3				0.905							
SS1					0.863						
SS2					0.964						
SS3					0.811						
PON1						0.612					
PON2						0.904					
PON3						1.002					
PCV1							0.800				
PCV2							0.973				
PCV3							0.673				
PS1								0.952			
PS2								0.843			
PS3								0.653			
TD1									0.545		
TD2									0.896		
TD3									0.873		
POP1										0.804	
POP2										0.946	
POP3										0.442	
PCT1											0.917
PCT2											0.704
PCT3											0.500

Extraction method: principal axis factoring. Rotation method: Promax with Kaiser normalization, *n* = 345. MTL = motivation for transfer learning; SSA = supervisor sanction; OUL = opportunity to use learning; LR = learning readiness; SS = supervisor support; PON = positive outcome negative; PCV = perceive content validity; PS = peer support; TD = training design; POP = personal outcome positive; PCT = personal capacity for transfer loading, and each variable had at least three components.

**Table 3 ijerph-19-09439-t003:** Rotated Factor Loading Table for the LTSI Training General Domain (5-Factor Extraction *n*
*=* 345).

	Factor				
	1	2	3	4	5
PSE1	0.698				
PSE2	0.724				
PSE3	0.985				
FPC1		0.919			
FPC2		0.887			
FPC3		0.731			
ROC1			0.724		
ROC2			0.845		
ROC3			0.889		
TEPE1				0.481	
TEPE2				0.895	
TEPE3				0.546	
TEPEx1					0.624
TEPEx2					0.613
TEPEx3					0.532

Extraction method: principal axis factoring. Rotation method: Promax with Kaiser normalization. *n* = 345. PSE = personal self-efficacy; FPC = feedback performance coaching; ROC = resistance/openness to change. TEPE = transfer effort performance expectation; TEPEx = transfer effort performance expectancy.

**Table 4 ijerph-19-09439-t004:** CFA Crosschecked Fit Statistic for 11 Factors for Training in the Specific Domain and 5 in the General Domain, in the Second Sample (*n* = 355).

Model	Chi-Square	Chi-Square /Df	CFI	TLI	RMSEA	RMSEA CI-Low	RMSEA CI-High	SRMR	Remarks
Optimal Model 1 from CFA1(a) (11 factors)	440	2.90	0.965	0.950	0.064	0.047	0.066	0.065	Good model fit
Optimal Model 1 from CFA1(b) (5 factors)	212.538	2.657	0.958	952	0.063	0.040	0.057	0.040	Good model fit

**Table 5 ijerph-19-09439-t005:** Reliability and AVE Convergent Validity of Training in the Specific Domain (*n* = 700).

	CR	AVE	MSV	MaxR(H)	OUL	SS	SSA	LR	PON	POP	PCV	PS	TD	MTL	PCT
OUL	0.957	0.881	0.167	1.019	0.939										
SS	0.945	0.852	0.394	0.947	0.071	0.923									
SSA	0.939	0.836	0.454	0.95	−0.089 *	0.004	0.915								
LR	0.93	0.817	0.153	0.935	0.022	0.221 ***	0.134 ***	0.904							
PON	0.938	0.834	0.207	0.959	0.06	0.169 ***	0.455 ***	0.391 ***	0.913						
POP	0.909	0.77	0.284	0.932	0.305 ***	0.308 ***	0.286 ***	0.186	0.186 ***	0.877					
PCV	0.916	0.785	0.493	0.93	0.273 ***	0.389 ***	−0.046	0.021	−0.098 *	0.163 ***	0.886				
PS	0.945	0.852	0.394	0.948	0.409 ***	0.627 ***	−0.024	0.228 ***	0.047	0.225 ***	0.437 ***	0.923			
TD	0.921	0.795	0.493	0.926	0.315 ***	0.361 ***	−0.073 *	0.115 ***	−0.012	0.184 ***	0.702 ***	0.436 ***	0.891		
MTL	0.923	0.801	0.386	0.924	0.384 ***	0.208 ***	0.013	0.045	−0.029	0.533 ***	0.527 ***	0.272 ***	0.621 ***	0.895	
PCT	0.936	0.829	0.454	0.947	0.231 ***	−0.095 *	0.674 ***	0.132 ***	0.425 ***	0.249 ***	0.350 ***	0.144 ***	0.225 ***	0.068 *	0.911
Cronbach’					0.951	0.941	0.938	0.929	0.936	0.904	0.915	0.945	0.92	0.923	0.935

* *p*< 0.05, *** *p* < 0.001 MTL = motivation to transfer learning; SSA = supervisor sanction; OUL = opportunity to use learning; LR = learning readiness; SS = supervisor support; PON = positive outcome negative; PCV = perceive content validity; PS = peer support; TD = training design; POP = personal outcome positive; PCT = personal capacity to transfer.

**Table 6 ijerph-19-09439-t006:** Reliability and AVE Convergent Validity of Training in the General Domain (*n* = 700).

	CR	AVE	MSV	MaxR(H)	FPC	ROC	PSE	TEPEx	TEPE
FPC	0.939	0.837	0.201	0.948	0.915				
ROC	0.939	0.838	0.063	0.959	0.251 ***	0.915			
PSE	0.919	0.792	0.512	0.941	0.206 ***	0.137 ***	0.89		
TEPEx	0.95	0.863	0.512	0.962	0.263 ***	0.157 ***	0.715 ***	0.929	
TEPE	0.902	0.756	0.375	0.929	0.448 ***	−0.072 *	0.563 ***	0.612 ***	0.87
Cronbach’					0.936	0.938	0.915	0.949	0.9

* *p*< 0.05, *** *p* <0.001 PSE = personal self-efficacy; FPC = feedback performance coaching; ROC = resistance/openness to change. TEPE = transfer effort performance expectation; TEPEx = transfer effort performance expectancy.

**Table 7 ijerph-19-09439-t007:** Model Fits for Invariant Test across Gender Training in the Specific Domain.

	χ^2^	Df	χ^2^/Df	Δχ^2^	ΔDF	*p*-Value	CFI	ΔCFI	RMSEA
Configural Invariance	1819.833	880	2.068				0.95		0.056
Metric invariance	1856.101	913	2.033	36.268	33	0.319	0.95	0	0.055
Scalar Invariance	1819.833	880	2.068	36.268	33	0.472	0.95	0	0.056

**Table 8 ijerph-19-09439-t008:** Model Fits for Invariant Test across Gender Training in the General Domain.

	χ^2^	Df	χ^2^/Df	Δχ^2^	ΔDF	*p*-Value	CFI	ΔCFI	RMSEA
Configural Invariance	320.336	160	2.002				0.954		0.054
Metric invariance	332.557	175	1.9	15	15	0.662	0.955	0.001	0.051
Scalar Invariance	332.557	175	175	0	0	0.955	0	0.955	0.051

**Table 9 ijerph-19-09439-t009:** Results of Difference Comparison.

	Difference of Latent Mean	Standard Deviations	Cohen’s d
Peer Support	0.15	1.06	0.14
Supervisors Sanction	0.17	1.20	0.14
Training Design	0.10	0.77	0.13

## Data Availability

The data that support the findings of this study are available from the corresponding author, B.N.C., upon reasonable request.

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
