# Peer review of "Psychometric Properties (Measurement Invariance and Latent Mean Difference across Gender) of the Learning Transfer Inventory System (LTIS) to Assess Thai Teachers’ Learning Transfer of COVID-19 Prevention Guidance in the Classroom"

_ijerph, 2022, doi:10.3390/ijerph19159439_

Round 1
Reviewer 1 Report
Dear Editor:
Thank you for this opportunity for me to review the manuscript titled “Psychometric Properties (Measurement Invariance and Latent Mean Difference Across Gender) of Learning Transfer Inventory System (LTIS) to Assess Thais Teachers Learning Transfer of COVID 19 Prevention Guidance in the Classroom.” I have now thoroughly read the manuscript and would like to share my opinions with you.
Overall, the manuscript is well-written and requires only a moderate level of editing in its language use. Though I might have asked for some clarifications, the methodology is well-described, and one should not have much difficulty following what has been done in this instrument validation study. However, I cannot recommend publication in your journal, not because of any methodological issues, but the lack of theoretical justification (thus, novelty and meaningfulness of the study). I will detail my concerns below.
1. The Introduction section provides background information for the study but does not discuss any specific measurement issues in measuring learning transfer. What is reason that a “most-widely used” scale as the LTIS still needs to be validated again? Just because it has been validated in the cultural context in “developing countries”? (L103). Then the discussion should have been focused on why a scale would perform differently in developing economies as versus developed, industrialized countries. But I don’t find such justifications.
2. Taking a closer look at the research questions or goals of the study (L112 to L118), one may notice that none of the goals hypotheses are well-supported by the literature review. There appears to be a disconnection between the introduction and the goals.
3. L121, “We recruited a non-probably . . .” should that be “non-probability” instead?
4. Please be consistent in your use of these terms: variable, factor, and dimension. Factors in factor analysis speak of the dimensionality of your scale. Does that mean the LTSI is a multidimensional scale with two higher-order factors: training specific and training general? If so, have the developers and previous validation studies tested this higher-order CFA? Also, do training specific and training general converge, meaning that users may create one composite score from all 48 items? Such is vital information, which should not be omitted in an instrument validation study.
5. What is the reason that EFA was conducted, if there was already an established factor structure of the LTSI? While I am by no means questioning the statistical procedure the researchers used in the EFA. It is rare for the items to load on their respective factors just as hypothesized, when the factor structure is so complex: a total of 16 factors that are correlated with each other. What method did the author use in determining the number of factors to extract? Is it possible that the researchers extracted more factors than they should? A related question is: how many items were used in the EFA? It says “48 items” (L.127) and then 33 items (L.207) in EFA. Where did the 15 items go? Dropped due to bad loadings and cross loadings (but then the researchers stated that there were no cross-loading items)?
6. What are the factor correlations like from CFA?
7. Measurement invariance. Again, we don’t do measurement invariance analysis just because we happen to have that variable named gender. There must be theoretical support for such investigations. Is there any reason for us to hypothesize that measurement biases exist between males and females? At least I do not find compelling arguments in the manuscript. Or, why gender? Why wasn’t measurement invariance examined across age groups, service experience groups, or in terms of degree (e.g., four-year college degree vs. others)? Their measurement invariance results in fact echoed my concern: the scale performed the same for males and females. By the way, the model fit information (delta chi-square, delta df, p value, CFI…) is certainly not right with the scalar invariance model (looks like they pasted the values to the wrong columns?)
8. Given that measurement invariance was upheld in all levels up to scalar invariance, the comparison of latent means between males and females is not substantially more meaningful than a paired-samples t test using the observed scores. Certainly, we cannot deny that latent means are better, more reliable measures than observed scores, but how much better? So much better that applied researchers should all derive latent scores from a factor analysis framework for statistical inferences? I do not see so. All Cohen’s ds indicate very small gender differences, in fact (see Table 9).
I applaud the researchers’ hard work, but still don’t see how this study might contribute to the knowledge base.
Reviewer 2 Report
Review for the manuscript “Psychometric Properties (Measurement Invariance and Latent Mean Difference Across Gender) of Learning Transfer Inventory System (LTIS) to Assess Thais Teachers Learning Transfer of COVID 19 Prevention Guidance in the Classroom.”
Date: 26.6.2022
Journal: IJERPH
Round: 1
This study aims to validate the LTIS instrument in Thailand. The research expands the validation of the instrument in a developing country. It has already been validated in various countries, but not developing that much.
All in all, the statistical side of the manuscript is very impressive, but I have few recommendations how to improve the work:
1. The title mentions psychometric properties, but they have not been defined or addressed in the theoretical framework.
2. On the line 157 there is an expression “flying colors”. To it is language used in daily discussion not in scientific papers.
3. Could you elaborate the supervisor sanction perspective on male. I am in an impression that in general women are more concerned from it than men. And this is due the cultural aspect of “boys will be boys”. Did I misunderstand something?
My last comment is about the sampling. For the validation a purposive sampling is very ok. I do not recommend repeating this work via random sampling. I consider this instrument validated, just start using it.
As a summary, this manuscript can be published after authors revise listed minor issues.
